## [Transparent Peer Review file · Nature Communications]

Sororin locks the DNA-exit gate of cohesin to preserve sister-chromatid cohesion

Corresponding Author: Professor Fangwei Wang

Version 0:

Reviewer comments:

Reviewer #1

(Remarks to the Author)

This is a well-designed and carefully executed paper that advances our understanding of Sororin's role in cohesion. The authors propose that Sororin's C-terminal region (CTR) directly locks the cohesin ring at the RAD21-SMC3 exit gate, beyond its known function as a Wapl inhibitor. The experiments are extensive, but many rely on overexpression of tagged mutants in HeLa or 293T cells and pull-down assays. While the data broadly support the model, the evidence for a "molecular lock" at the Smc3-Rad21 gate is still indirect. The conclusion that Sororin's CTR acts independently of SA2 is also difficult to reconcile with earlier reports showing Sororin-SA2 interaction. A more nuanced view may be that Sororin interacts with SA2/Pds5 during initial loading or stabilization and then switches to Rad21-Smc3 after cohesin acetylation to secure/strengthen the exit gate. This sequential or context-dependent interaction should be considered and could be tested, for example, with the S145A Sororin mutant.

The data redefine Sororin as an "autonomous" cohesin lock, yet Sororin's function is still initiated by Escs1/2-dependent recruitment and cooperation with Pds5. Emphasizing how Sororin's gate engagement is spatiotemporally restricted (only becoming critical at centromeres or after cohesin acetylation) would help readers grasp the mechanistic innovation. Finally, some methodological aspects warrant scrutiny. The artificial tethering strategies (centromeric and chromatin-wide) are powerful tests of sufficiency, but they represent non-physiological scenarios. For instance, chromatin-wide CTR tethering phenocopies Wapl loss with severe condensation and segregation defects, which supports the model but might have off-target effects (e.g. steric hindrance of other chromatin processes). The centromere-targeting rescue experiment is convincing, yet only partially compensates for Sgo1 loss, suggesting Sororin-CTR alone cannot fully substitute all centromeric cohesion safeguards - this nuance should be discussed. In general, the experimental evidence does support the central claims that Sororin's CTR is necessary and sufficient to stabilize cohesion by counteracting Wapl-mediated release; however, the authors should address these points to ensure the conclusions are fully justified by the data. In its current form, the work is technically sound and quite novel, but bolstering the direct "lock" evidence and clarifying the model would align it better with Nature Communications' standards of rigor and conceptual insight.

Figure-specific comments:

Fig. 2B: The claim of 100% Sororin knockdown is not realistic; provide the full blot.

Fig. 2G,H: Show siRNA depletion efficiency for Escs1/2.

Fig. 4: Can Sororin pull down SA1 in SA2 KO cells? If so, how is this distinguished from the essential Rad21-CTR interaction? Using SA2-deficient cancer lines would strengthen the case.

This reviewer remain unconvinced that a Rad21/Smc3 "sub-complex" exists physiologically. This is an intriguing possibility but requires direct verification.

Fig. 4A: A full-length Sororin control is missing. Sororin-CTR pulls down Smc1/3, Rad21, SA2 but not Pds5B; Sororin 1-222 pulls down only Pds5B. Can full-length Sororin immunoprecipitate all cohesin components?

Fig. 4F: GST-Sororin-CTR pulls down overexpressed Smc3-GFP/Rad21-Myc efficiently, but this may artificially generate a Rad21/Smc3 sub-complex. The authors should repeat with non-overexpressed lysates and probe the stoichiometry of Smc1, Smc3, and Rad21 to show whether genuine sub-complexes exist.

Fig. 5: Can the S145A Sororin mutant still bind Rad21?

Rigor and reproducibility: All functional data are limited to HeLa or 293T cells, with no validation in non-transformed or additional lines. The SMC3(KE/KE) experiment shows only a mild phenotype, likely due to viability issues, which weakens the argument for the SMC3 surface. A conditional degron or dTAG system would allow a cleaner test at physiological levels.

Similarly, the extent of Wapl suppression across different mutants (Sororin, Rad21, Smc3) should be quantified systematically, and Wapl overexpression tested. It would also be useful to examine Pds5A versus Pds5B specificity. Finally, the manuscript is very dense. Simplifying terminology and emphasizing the take-home message would help readers. For instance, the term “gate engagement” is used repeatedly-defining it clearly up front and explaining how Sororin’s gate-locking complements Wapl/Pds5 regulation would prevent confusion.

Reviewer #2

(Remarks to the Author)

The manuscript by Chen et al. presents a compelling and mechanistically novel function for Sororin in directly locking the RAD21–SMC3 interface of the cohesin ring, thereby acting as a guardian of the DNA exit gate to preserve sister-chromatid cohesion. The extreme C-terminal region (CTR) of Sororin is essential for the maintenance of sister chromatid cohesion. Artificial tethering of the Sororin CTR to chromosome centromeres can rescue cohesion in Sororin-depleted cells, while genome-wide tethering phenocopies Wapl loss, causing abnormal chromosome condensation, decatenation, and segregation. To further investigate the molecular mechanism of Sororin’s function, the authors combine AlphaFold3-guided modeling with a series of biochemical assays and identify key conserved residues in Sororin, RAD21, and SMC3 that mediate this interaction and demonstrate their functional necessity for cohesion. Furthermore, they dissect the regulation of Sororin, showing that mitotic phosphorylation by Aurora B at S145 selectively disrupts its interaction with Pds5 but not with the RAD21-SMC3 interface. Finally, they demonstrate that the cohesion defects caused by disrupting the Sororin-RAD21-SMC3 interaction are entirely Wapl-dependent. The authors therefore propose a dual-function model where Sororin acts both as a competitive displacer of Wapl (via Pds5) and as a direct structural lock for the cohesin exit gate. demonstrated that Sororin directly binds to the SMC3–RAD21 interface.

From these lines of evidence, the authors conclude that Sororin’s CTR directly binds the SMC3–RAD21 interface to seal cohesin’s DNA exit gate. This sealing activity works synergistically with Pds5 competition to balance sister chromatid cohesion maintenance and regulated dissociation. Overall, the in vitro and in vivo data are clear and support the conclusions. This work provides a comprehensive understanding of how Sororin functions in cohesion and highlights the sequential and nonredundant roles of SMC3 acetylation and Sororin gatelocking: acetylation licenses Sororin recruitment, while Sororin–CTR engagement locks the exit gate. These findings will be of general interest to the fields of chromosome biology and cell cycle regulation. I enthusiastically recommend publication pending the addressing of the points below.

Major points:

1. In Figure 2, acetyltransferases Esco1 and Esco2 acetylate K105 and K106 in the ATPase head of SMC3 (SMC3-HD) during S phase, which promotes Sororin recruitment to stabilize cohesin on chromatin. Overexpression of the Sororin CTR fails to rescue cohesion defects caused by ESCO1/2 knockdown. The authors reason that ESCO1/2-mediated acetylation of SMC3 licenses Sororin recruitment, and that engagement of Sororin-CTR subsequently locks the cohesin exit gate. Does SMC3 acetylation enhance the interaction between Sororin and the SMC3–RAD21 interface? SMC3 interacts with Sororin through K1034, K1038, and R1099. Do these amino acids spatially localize near the acetylation sites of SMC3? It may be worth testing whether acetylation-deficient (K105A/K106A) and acetylation-mimetic (K105Q/K106Q) mutations in SMC3-HD affect the binding of the SMC3-HD/Rad21-N103 subcomplex to Sororin-CTR and/or full-length Sororin.
2. In Figure 2 (G, H), tethering Sororin-CTR to centromeres had little effect on cohesion in cells co-depleted of Esco1/2. Does Sororin-CTR affect its binding to chromatin at the cohesion establishment stage? It would be interesting to test whether expression of H2B-fused Sororin-CTR can rescue cohesion defects in Esco1/2-depleted cells.
3. Sororin is phosphorylated by mitotic kinases PLK1, CDK1, and Aurora B in vivo, which mediates its degradation. The authors found that Aurora B phosphorylation of Sororin at S145 reduces its interaction with Pds5 in vitro. Is this the case in vivo? What’s the order of these phosphorylation events which may mediate dissociation from PDS5 and degradation sequentially?
4. Sgo1 plays a well-established role in sustaining Sororin activity at mitotic centromeres. Figure S2 (A-C) shows that tethering Sororin-CTR to centromeres, as a fusion protein with a CENP-B fragment, only partially restored cohesion defects caused by Sgo1 depletion. This is somewhat striking, since Sgo1 is generally regarded as a master protector of centromeric cohesion. This result suggests that Sgo1 may have an additional role in protecting cohesion along mitotic chromosome arms. To test this possibility, the authors should assess the effect of chromatin-tethered Sororin-CTR on cohesion in Sgo1-depleted cells.
5. Figure 3H shows that, compared to H2B-GFP-expressing cells, the frequency of anaphases with chromosome bridges was strongly increased in H2B-Sororin-CTR-expressing cells, whereas the lagging chromosome rates remained comparable. Given the critical role of centromeric Aurora B kinase in correcting erroneous kinetochore-microtubule attachments, the authors should compare Aurora B accumulation at mitotic centromeres in cells expressing H2B-GFP versus H2B-Sororin-CTR-GFP.
6. Figures 4G and S3 show that GST-Sororin-CTR interacted with SMC3-HD only when it formed a subcomplex with the N-terminal fragment of RAD21. This suggests that Sororin-CTR binds the cohesin core complex in which the DNA-exit gate (i.e, the SMC3-HD/RAD21-N interface) is closed, but not when the gate is open. Because cohesin-mediated sister chromatid cohesion is established during S phase, maintained in G2, and largely lost in mitosis, the authors should compare the relative binding efficiency of Sororin-CTR to endogenous cohesin core complex in early S phase versus G2, and in G2 versus mitosis.
7. Using bacterially expressed and purified proteins, Figure 4H shows that GST-Sororin-CTR pulled down the complex formed between RAD21-N103 and SMC3-HD. This indicates a direct interaction between Sororin-CTR and the RAD21/SMC3-HD interface. To corroborate this, the authors should test whether the RAD21-N103/SMC3-HD complex can directly pull down GST-fused full-length Sororin.
8. It is well known that Sororin phosphorylation during mitosis promotes cohesin release from chromosome arms. Figure 5A shows that Sororin efficiently co-immunoprecipitated RAD21-N103, SMC3-HD, and endogenous Pds5B in G2-phase cell. In

mitosis, Sororin binding to RAD21-N103 and SMC3-HD persisted, while the Sororin-Pds5B interaction was disrupted. Figure 5B demonstrates that co-inhibition of Cdk/Plk1/Aurora kinase activity during mitosis restored the Sororin-Pds5B interaction but did not affect Sororin binding to RAD21-N103/SMC3-HD. These results imply that Sororin-CTR binding to the RAD21/SMC3-HD interface is insensitive to mitotic phosphorylation, in contrast to the Sororin-Pds5B interaction. To strengthen this conclusion, the authors should examine the effect of treating mitotic cell lysates with lambda-phosphatase on Sororin binding to RAD21-N103/SMC3-HD and Pds5B.

9. Figure 6 (C, D) shows simultaneous mutation of W233 and E252 to alanine and lysine, respectively disrupts Sororin-CTR binding to the SMC3-HD/RAD21-N103 subcomplex and to endogenous cohesin. A Sororin-E252Q mutation is reported in the cancer genome database. It would be interesting to see to what extent this mutation can affect the binding of full-length Sororin to SMC3 and RAD21 in cells.

Minor points:

1. The statistical results lack critical information: such as what the values represent (\pm SD or \pm SEM), the method of significance analysis, the corresponding p-values, and the number of biological replicates. Fig. 1D and 1G; Fig. 2D and 2E lack significance analysis. Fig. 3G lacks error bars, and no significance analysis was performed for Fig. 3H and 3I. Fig. 7B, 7G, 7I; Fig. 8I; Fig. 9B, 9E lack significance analysis.
2. The Methods section states that statistical analyses were performed using GraphPad Prism version 6, but do not specify the exact tests used (e.g., unpaired t-test, ANOVA). This should be clarified.
3. Could the authors provide the 3D structure of the cohesin complex or protein-protein interactions predicted by AlphaFold3?
4. Some grammars and typos, for instance: "phenocopying Wapl loss" in the Abstract better to be changed to "phenocopying the effects of Wapl loss". "Dysregulation of cohesin underlie developmental disorders and contribute to tumorigenesis" in the Introduction section should be changed to "Dysregulation of cohesin underlies developmental disorders and contributes to tumorigenesis". "Wapl regulates the dynamic association of cohesin with chromatin" in the Results section better to be changed to "Wapl regulates the dynamics of cohesin's association with chromatin". "To rule out the possibility that the failure of Sororin (1-149)GFP and Sororin (1-222)GFP to promote cohesion was due to deficient chromatin binding" in the Results section better to be changed to "To rule out the possibility that the failure of Sororin (1-149)GFP and Sororin (1-222)GFP to promote cohesion was due to defective chromatin binding".

Reviewer #3

(Remarks to the Author)

The manuscript by Chen et al. investigates the mechanisms by which Sororin functions to stabilize cohesive cohesin on DNA. While Sororin's antagonistic role against WAPL-mediated cohesin removal has been recognized, the underlying mechanisms remained poorly defined. Previous models proposed that Sororin and WAPL compete for PDS5 binding, however, this study reveals a different mechanism in which, in addition to interacting with PDS5, Sororin directly interacts with the cohesin complex, functioning as a gatekeeper for the SMC3-RAD21 exit gate. The authors demonstrate that the C-terminal domain of Sororin is sufficient to maintain sister chromatid cohesion and that tethering of a C-terminal Sororin construct to centromeres functionally protects sister chromatid cohesion and counters WAPL-mediated removal. Despite this protective effect, knockdown of ESCO1/2 still results in cohesion defects even when Sororin is tethered to centromeres, indicating that cohesin acetylation occurs upstream of Sororin binding and that stabilization of Cohesin by acetylation is required independently to maintain cohesion. Importantly, when Sororin is tethered along the chromosome arms and cannot be removed, it leads to mitotic segregation defects and abnormal chromosome compaction, indicating the importance of dynamic regulation and maintenance of cohesive cohesin at centromeres but not at chromosome arms in mitosis. Through immunoprecipitation experiments and elegant structure-function analyses, the authors demonstrate that Sororin directly interacts with cohesin and define the specific interaction interfaces. Using both in vitro and in vivo binding assays, they map the binding interfaces at the SMC3-RAD21 junction and validate these through reciprocal mutagenesis. Critically, co-depletion of both WAPL and Sororin followed by expression of binding mutants restores cohesion, confirming that exit gate locking is required specifically to block WAPL-mediated unlocking rather than to maintain cohesion directly. In addition to its interaction with cohesin, the authors find that Sororin binds PDS5B in an Aurora phosphorylation-dependent manner, identifying the specific phosphorylation residue and testing this through both in vitro and cell-based kinase assays. Interestingly, the phosphorylation of Sororin by Aurora regulates Sororin binding to PDS5B but not to cohesin, indicating that the association of Sororin with cohesin and PDS5 is independent, although the Sororin-cohesin interaction was reduced. Together, this work provides a comprehensive answer to the long-standing question of how Sororin counteracts WAPL-mediated cohesin removal. The authors employ multiple complementary approaches to test their model and draw balanced conclusions from their data. Given its relevance to genome regulation, human health, and its potential to advance the cohesin field, I recommend this manuscript for publication in Nature Communications. I have listed a few minor suggestions below that could strengthen the manuscript.

Suggestions:

1. The authors propose a mechanism in which phosphorylation of Sororin by AURKB releases the Sororin-PDS5 interaction while maintaining the Sororin-cohesin interaction, providing a switch whereby arm cohesin (but not centromeric cohesin) can be released during early mitosis. They suggest that arm cohesin has less Sororin CTR engagement due to lowered Sororin levels, allowing for exit gate opening. However, I find this model somewhat confusing: if Sororin is still interacting with PDS5 despite lowered Sororin levels, why would the CTR of these same Sororin proteins not also bind to cohesin in these same complexes? An alternative interpretation is that the pull-down of Sororin with cohesin in the ASE mutants reflects Sororin bound to soluble cohesin that is being removed from chromatin, rather than chromatin-bound cohesive cohesin. Typically, Sororin only interacts with chromatin-bound cohesin except in cases where cohesin cannot be acetylated; however, it is possible the authors are capturing this transient interaction when cohesin is being released. To help support the authors' proposed model, it would be useful to perform the IP experiments coupled with cellular

fractionation. Additionally, it would be useful to include chromosome spreads in the ASE mutant in the absence of endogenous Sororin to show that cohesive cohesin remains intact specifically at centromeres but not at chromosome arms. However, it is appreciated that these are complicated experiments, and it would be appropriate to alternatively mention other possible hypotheses in the discussion.

2. The authors find that tethering of Sororin to centromeres can only partially rescue Shugoshin knockdown, consistent with a model in which Shugoshin-mediated PP2A dephosphorylation blocks the release of the Sororin-PDS5 interaction at centromeres, indicating both interaction with PDS5 and cohesin is required to protect cohesin from premature removal. To further test this regulatory pathway, can a non-phosphorylatable Sororin mutant bypass the requirement for Shugoshin protection, functioning to block Aurora phosphorylation? This experiment would provide additional validation of the proposed phosphorylation-dependent switch mechanism and the need for both sororin interaction interfaces to mediate cohesive cohesin.

3. It would be useful to add statistics to the figures throughout.

4. Typo line 129, "whether selectively retention".

Reviewer #4

(Remarks to the Author)

How sister chromatid cohesion is established and maintained is a key question in the field of chromosome segregation. Extensive efforts from many labs have identified the major players in these processes and revealed a lot of basics about how these players work together to regulate sister chromatid cohesion. Among these players are Wapl, Pds5 and Sororin. It has been believed that the positive factor Sororin competes against factor Wapl for Pds5 binding, thus dictating "on and off" of sister chromatid cohesion. It has also been shown that the C-terminal domain of Sororin is required for sister chromatid cohesion; but the underlying mechanism is unknown. In addition, it remains elusive how these proposed mechanisms work together to regulate sister chromatid cohesion. Very interestingly, Chen et.al in this study found that the competition between Wapl and Sororin on Pds5 may not be that important for the regulation of sister chromatid cohesion. Instead, they revealed a novel mechanism, in which Sororin engage its C-terminal domain to directly binds to "the gate" of the cohesin complex that forms between Smc3 and Rad21, which competes against the action of Wapl, thus protecting sister chromatid cohesion. The major conclusions in this paper are solid and largely supported by plenty of data. These conclusions will contribute for us to further understand the molecular mechanisms of sister chromatid cohesion. The experiments were well controlled and carried out. And the paper was well organized and written. Especially, a series of point mutants in cohesin and Sororin identified in this study significantly strengthen their conclusion. I congratulate the authors for this great study. I do not have any major concerns and support the publication in NC

Minor points:

1. The authors already demonstrated that the expression of the two C-terminal domains are able to close the chromosome arm. Especially, for the smaller C-terminal domain (223-252), I believe that most of regulatory domains or residues in Sororin have been deleted out, for example, the Cdk1 and Aurora B phosphorylation sites. Whether this fragment will always bind Smc3/Scc1 throughout the cell cycle? Does it also generate anaphase bridges just like the H2B-fused fragment? It would be nice if author have the data. Otherwise, a brief discussion would also be fine.

Reviewer #5

(Remarks to the Author)

Version 1:

Reviewer comments:

Reviewer #1

(Remarks to the Author)

The authors have addressed my concerns with significant new data that substantially improve the rigor of the manuscript. I have a few general comments. The figure legends require further improvement so that readers can fully appreciate and interpret the data without having to refer back to the Results section. For example, in Fig. 4M (top two panels), "Sororin-GFP," "S." and "L." should be explicitly defined in the legend rather than left to assumption.

Reviewer #2

(Remarks to the Author)

All comments have been addressed with new data. It's now solid and ready for publication.

Reviewer #3

(Remarks to the Author)

The manuscript by Chen et al. addresses a long standing question of how Sororin acts to counteract WAPI based cohesin removal for the establishment of sister chromatid cohesion. In this new revised version of the manuscript the authors have added a number of experiments, included new analysis and statistics, and importantly added in additional discussion points that together has significantly strengthened the work. With these changes all of our concerns have been addressed and we would recommend this manuscript for publication in Nature Communications in its current form.

Reviewer #4

(Remarks to the Author)

I am satisfied with the authors' response.

Reviewer #5

(Remarks to the Author)

Point-to-point response to reviewer comments

We sincerely thank all reviewers for their insightful and constructive comments, which we have now fully addressed. All the important changes in the revised manuscript are underlined. Please find our detailed, point-by-point responses to the reviewers' comments below (in blue).

REVIEWER COMMENTS

Reviewer #1 (Remarks to the Author):

This is a well-designed and carefully executed paper that advances our understanding of Sororin's role in cohesion. The authors propose that Sororin's C-terminal region (CTR) directly locks the cohesin ring at the RAD21-SMC3 exit gate, beyond its known function as a Wapl inhibitor. The experiments are extensive, but many rely on overexpression of tagged mutants in HeLa or 293T cells and pull-down assays. While the data broadly support the model, the evidence for a "molecular lock" at the Smc3-Rad21 gate is still indirect.

Response: We thank for the thoughtful and constructive feedback, which has been instrumental in strengthening our manuscript. We appreciate the reviewer for acknowledging the study's design and execution, and for the supportive comment regarding our model.

We agree that strengthening the direct evidence for our model was essential. In response to this and other reviewers' comments, we have significantly expanded our experimental evidence beyond overexpression and pull-down assays:

1) Chromatin-specific interaction: New chromatin fractionation experiments demonstrate that Sororin-GFP co-immunoprecipitates RAD21, SMC3, and Pds5B specifically from the chromatin-bound fraction of mitotic cells, but not from the soluble fraction (Fig. 4M, N; **new data**). This confirms that our detected interactions are physiologically relevant to chromatin-associated cohesin.

2) Direct binding with recombinant proteins: We now show that the recombinant RAD21-N103/SMC3-HD complex pulls down full-length GST-Sororin (Fig. 4J; **new data**), corroborating the direct nature of the interaction. Moreover, we demonstrate that GST-Sororin-CTR specifically pulls down the recombinant RAD21-N103/SMC3-HD complex (Fig. 4I), and that mutations of residues mediating Sororin-CTR binding to the RAD21/SMC3 interface disrupt the direct interaction of Sororin-CTR with the RAD21/SMC3 complex (Fig. 6C, H).

3) Cell-cycle regulated interaction: New data reveal that Sororin-CTR binding to cohesin is highly cell-cycle-dependent, being strongest in G2 phase and weak in early S phase and mitosis (Fig. 4K, L; **new data**), aligning perfectly with the window of

cohesion maintenance.

4) RNAi-rescue assays: By replacing endogenous Sororin (Fig. 7F-J), RAD21 (Fig. 7C-E), and SMC3 (Fig. 7H-J; **new data**) with exogenously expressed proteins, we show that the direct binding of Sororin-CTR to the RAD21/SMC3 interface is important for the protection of sister-chromatid cohesion in cells.

5) Validation in non-transformed cells: Key findings have been validated in hTERT-RPE1 cells, demonstrating that the cohesion-stabilizing effect of Sororin-CTR and the defect of its binding mutants are not cell-line-specific (Supplementary Fig. 7B-D, H-M; **new data**).

The conclusion that Sororin's CTR acts independently of SA2 is also difficult to reconcile with earlier reports showing Sororin-SA2 interaction. A more nuanced view may be that Sororin interacts with SA2/Pds5 during initial loading or stabilization and then switches to Rad21-Smc3 after cohesin acetylation to secure/strengthen the exit gate. This sequential or context-dependent interaction should be considered and could be tested, for example, with the S145A Sororin mutant.

Response: We appreciate this insightful suggestion for a more nuanced model.

It is important to note that the direct Sororin-SA2 interaction was reported in a single study (Zhang N. & Pati D., Cell Cycle, 2015, PMID: 25608232) and has not been widely substantiated in the field. Our new data indicate that the reported Sororin-SA2 interaction is mediated indirectly through RAD21:

1) Sororin-CTR retains binding to SMC1, SMC3, and RAD21 in SA2-knockout cells (Fig. 4C, D);

2) The SA2-D793K mutation, which disrupts its binding to RAD21, also abolishes its interaction with Sororin-CTR (Supplementary Fig. 4C, D; **new data**).

As suggested, we have integrated a more nuanced model into the revised Discussion. Our new data show that the phospho-deficient Sororin-S145N mutant increases association with Pds5B, SMC3, and RAD21 specifically in mitosis but not in G2 phase (Supplementary Fig. 5F, G; **new data**). This supports a model where Sororin first engages Pds5 during cohesion establishment and, following cohesin acetylation, its CTR locks the SMC3-RAD21 interface for stabilization (revised Fig. 9J). Given that the Pds5-binding-deficient Sororin-ASE mutant still associates with cohesin core subunits, albeit less efficiently (Fig. 5D, E), we clarify that while the Pds5 interaction contributes to Sororin's association, the CTR-mediated gate lock is essential and can function independently.

The data redefine Sororin as an "autonomous" cohesin lock, yet Sororin's function is still initiated by Escs1/2-dependent recruitment and cooperation with Pds5. Emphasizing how Sororin's gate engagement is spatiotemporally restricted (only becoming critical at centromeres or after cohesin acetylation) would help readers grasp the mechanistic innovation.

Response: We thank the reviewer for this excellent suggestion to highlight mechanistic innovation. We have added new data and explicit discussion on this point:

1) Temporal regulation: As mentioned above, new experiments demonstrate cell cycle-specific binding of Sororin-CTR to cohesin (Fig. 4K; **new data**);

2) Spatial regulation: Sororin-GFP co-immunoprecipitates RAD21, SMC3, and Pds5B specifically from the chromatin fraction, but not the soluble fraction, of mitotic cells (Fig. 4M, N; **new data**); moreover, centromere-tethered Sororin-CTR bypasses the requirement for endogenous Sororin in cohesion protection (Fig. 2B-F).

We now explicitly discuss this regulated, context-dependent engagement in the revised Discussion, clarifying how Sororin's role as a lock is deployed specifically at the right time and place.

Finally, some methodological aspects warrant scrutiny. The artificial tethering strategies (centromeric and chromatin-wide) are powerful tests of sufficiency, but they represent non-physiological scenarios. For instance, chromatin-wide CTR tethering phenocopies Wapl loss with severe condensation and segregation defects, which supports the model but might have off-target effects (e.g. steric hindrance of other chromatin processes). The centromere-targeting rescue experiment is convincing, yet only partially compensates for Sgo1 loss, suggesting Sororin-CTR alone cannot fully substitute all centromeric cohesion safeguards - this nuance should be discussed. In general, the experimental evidence does support the central claims that Sororin's CTR is necessary and sufficient to stabilize cohesion by counteracting Wapl-mediated release; however, the authors should address these points to ensure the conclusions are fully justified by the data. In its current form, the work is technically sound and quite novel, but bolstering the direct "lock" evidence and clarifying the model would align it better with Nature Communications' standards of rigor and conceptual insight.

Response: We thank the reviewer for appreciating that our evidence supports the central claims and that the work is technically sound and novel.

Regarding the potential off-target effects of chromatin tethering, we agree that this is an important consideration. To rule out non-specific steric hindrance, we show that tethering mutants of H2B-Sororin or H2B-Sororin-CTR (FA/FA or WA/EK) that are defective in binding the SMC3-RAD21 interface do not cause chromosome condensation or segregation defects (Fig. 7A-C; Supplementary Fig. 7A-J with **new panels of B-D and H-J**). The phenotype is therefore specific to the gate-locking function.

Regarding the partial rescue of cohesion in Sgo1-depleted cells by centromere-tethered Sororin-CTR, we have added the suggested nuance. In the revised manuscript, we state that "This partial rescue suggests that while centromeric Sororin-CTR can mitigate Sgo1 depletion, it cannot fully substitute for Sgo1, which also protects cohesin through PP2A-mediated dephosphorylation". Furthermore, new data show that constitutive chromatin tethering of Sororin-CTR largely bypasses the need for Sgo1 (Supplementary Fig. 3A, B; **new data**). We therefore state in the revised

manuscript that “This indicates that chromatin-tethered Sororin-CTR can largely bypass the requirement for Sgo1, supporting the model that Sgo1 protects cohesin both at centromeres and on chromosome arms”.

Figure-specific comments:

Fig. 2B: The claim of 100% Sororin knockdown is not realistic; provide the full blot.

Response: Corrected. The revised Fig. 2B now shows full blots with longer exposure, confirming that knockdown, while efficient, is not 100%.

Fig. 2G,H: Show siRNA depletion efficiency for Esco1/2.

Response: Added. Depletion efficiency for Esco1 is shown by immunoblotting (Fig. 2I; **new data**). For Esco2, due to a lack of a suitable antibody, we demonstrate knockdown via RT-PCR (Fig. 2J; **new data**).

Fig. 4: Can Sororin pull down SA1 in SA2 KO cells? If so, how is this distinguished from the essential Rad21-CTR interaction? Using SA2-deficient cancer lines would strengthen the case.

Response: Added. GST-Sororin-CTR retains the ability to bind SA1 in SA2-knockout cells (Supplementary Fig. 4A, B; **new data**). Since both SA1 and SA2 bind RAD21 (Sumara I. et al., J Cell Biol, 2000, PMID: 11076961; Canudas S. & Smith S., J Cell Biol, 2009, PMID: 19822671), we propose that Sororin-CTR pulls down SA1 indirectly via RAD21, consistent with our finding that the Sororin-SA2 interaction is RAD21-dependent.

This reviewer remain unconvinced that a Rad21/Smc3 “sub-complex” exists physiologically. This is an intriguing possibility but requires direct verification.

Response: We apologize for the misleading terminology. We do not propose a stable, free RAD21-SMC3 sub-complex. The term has been replaced throughout the revised manuscript with “RAD21-SMC3 interface” to accurately reflect our model of Sororin binding to their interaction surface.

Fig. 4A: A full-length Sororin control is missing. Sororin-CTR pulls down Smc1/3, Rad21, SA2 but not Pds5B; Sororin 1–222 pulls down only Pds5B. Can full-length Sororin immunoprecipitate all cohesin components?

Response: Added. New data show that full-length GST-Sororin pulls down endogenous Pds5B, SMC3, SMC1, RAD21, and SA2 (Fig. 4A; **new data**). Furthermore, full-length Sororin-GFP co-immunoprecipitates these cohesin components (Fig. 5D, 7E).

Fig. 4F: GST–Sororin-CTR pulls down overexpressed Smc3-GFP/Rad21-Myc efficiently, but this may artificially generate a Rad21/Smc3 sub-complex. The authors should repeat with non-overexpressed lysates and probe the stoichiometry of Smc1, Smc3, and Rad21 to show whether genuine sub-complexes exist.

Response: As clarified above, our model does not involve a stable RAD21-SMC3

sub-complex. The experiment demonstrates Sororin-CTR's ability to bind the RAD21-SMC3 interface. We have revised the text to avoid any implication of a pre-formed sub-complex.

Fig. 5: Can the S145A Sororin mutant still bind Rad21?

Response: Added. New co-immunoprecipitation data show that the phospho-deficient Sororin-S145N mutant exhibits increased association with Pds5B, SMC3, and RAD21 specifically in mitosis but not in G2 phase (Supplementary Fig. 5F, G; **new data**).

Rigor and reproducibility: All functional data are limited to HeLa or 293T cells, with no validation in non-transformed or additional lines.

Response: We now validate key findings in non-transformed RPE-1 cells:

1) H2B-Sororin-CTR-GFP (H2B-Sororin-GFP), but not the WA/EK or FA/FA mutants, prevented mitotic chromosome arm resolution (Supplementary Fig. 7B-D, H-J; **new data**);

2) Exogenous wild-type Sororin-SFB, but not the WA/EK or FA/FA mutants, rescues cohesion loss after endogenous Sororin depletion (Supplementary Fig. 7K-M; **new data**).

The SMC3(KE/KE) experiment shows only a mild phenotype, likely due to viability issues, which weakens the argument for the SMC3 surface. A conditional degron or dTAG system would allow a cleaner test at physiological levels. Similarly, the extent of Wapl suppression across different mutants (Sororin, Rad21, Smc3) should be quantified systematically, and Wapl overexpression tested.

Response: We have addressed these points:

1) After optimizing siRNA transfection to minimize cell death, SMC3 knockdown in HeLa cells now yields a clearer cohesion defect (Fig. 8H-J; **new data**). We state in the revised manuscript that "Depletion of endogenous SMC3 in these cells caused a cohesion defect in control cells (Fig. 8H-J), albeit milder than that observed following RAD21 depletion, likely due to reduced cell viability upon SMC3 loss. Notably, this defect was rescued by expression of wild-type SMC3-GFP but not by the SMC3-KE/KE-GFP mutant, indicating that SMC3 residues mediating Sororin-CTR binding are required for proper cohesion".

2) We provide new quantification showing that Wapl depletion prevents cohesion loss in cells where endogenous SMC3 is replaced with the Sororin-binding-deficient SMC3-KE/KE-GFP mutant (Fig. 9G-I; **new data**).

It would also be useful to examine Pds5A versus Pds5B specificity.

Response: Added. New data show that GST-Sororin, but not the GST-Sororin-ASE mutant, pulls down endogenous Pds5B (but not its paralog Pds5A) from HeLa cell lysates (Fig. 5E).

Finally, the manuscript is very dense. Simplifying terminology and emphasizing the

take-home message would help readers. For instance, the term “gate engagement” is used repeatedly—defining it clearly up front and explaining how Sororin’s gate-locking complements Wapl/Pds5 regulation would prevent confusion.

Response: We thank the reviewer for this suggestion. We have streamlined terminology, clarified key concepts like “gate engagement” upon first use, and emphasized the core message throughout the revised text to improve readability and conceptual clarity.

We believe these comprehensive revisions, which include substantial new data, have significantly bolstered the direct evidence for our model, clarified its nuances, and addressed all methodological concerns.

Reviewer #2 (Remarks to the Author):

The manuscript by Chen et al. presents a compelling and mechanistically novel function for Sororin in directly locking the RAD21–SMC3 interface of the cohesin ring, thereby acting as a guardian of the DNA exit gate to preserve sister-chromatid cohesion. The extreme C-terminal region (CTR) of Sororin is essential for the maintenance of sister chromatid cohesion. Artificial tethering of the Sororin CTR to chromosome centromeres can rescue cohesion in Sororin-depleted cells, while genome-wide tethering phenocopies Wapl loss, causing abnormal chromosome condensation, decatenation, and segregation. To further investigate the molecular mechanism of Sororin’s function, the authors combine AlphaFold3-guided modeling with a series of biochemical assays and identify key conserved residues in Sororin, RAD21, and SMC3 that mediate this interaction and demonstrate their functional necessity for cohesion. Furthermore, they dissect the regulation of Sororin, showing that mitotic phosphorylation by Aurora B at S145 selectively disrupts its interaction with Pds5 but not with the RAD21-SMC3 interface. Finally, they demonstrate that the cohesion defects caused by disrupting the Sororin-RAD21-SMC3 interaction are entirely Wapl-dependent. The authors therefore propose a dual-function model where Sororin acts both as a competitive displacer of Wapl (via Pds5) and as a direct structural lock for the cohesin exit gate. demonstrated that Sororin directly binds to the SMC3–RAD21 interface.

From these lines of evidence, the authors conclude that Sororin’s CTR directly binds the SMC3–RAD21 interface to seal cohesin’s DNA exit gate. This sealing activity works synergistically with Pds5 competition to balance sister chromatid cohesion maintenance and regulated dissociation. Overall, the in vitro and in vivo data are clear and support the conclusions. This work provides a comprehensive understanding of how Sororin functions in cohesion and highlights the sequential and non-redundant roles of SMC3 acetylation and Sororin gate-locking: acetylation licenses Sororin recruitment, while Sororin–CTR engagement locks the exit gate. These findings will be of general interest to the fields of chromosome biology and cell cycle regulation. I enthusiastically recommend publication pending the addressing of the points below.

Response: We thank the reviewer for their enthusiastic support and insightful suggestions. Below are our point-by-point responses and a summary of the corresponding revisions made to the manuscript.

Major points:

1. In Figure 2, acetyltransferases Esco1 and Esco2 acetylate K105 and K106 in the ATPase head of SMC3 (SMC3-HD) during S phase, which promotes Sororin recruitment to stabilize cohesin on chromatin. Overexpression of the Sororin CTR fails to rescue cohesion defects caused by ESCO1/2 knockdown. The authors reason that ESCO1/2-mediated acetylation of SMC3 licenses Sororin recruitment, and that engagement of Sororin-CTR subsequently locks the cohesin exit gate. Does SMC3 acetylation enhance the interaction between Sororin and the SMC3–RAD21 interface? SMC3 interacts with Sororin through K1034, K1038, and R1099. Do these amino acids spatially localize near the acetylation sites of SMC3? It may be worth testing whether acetylation-deficient (K105A/K106A) and acetylation-mimetic (K105Q/K106Q) mutations in SMC3-HD affect the binding of the SMC3-HD/Rad21-N103 subcomplex to Sororin-CTR and/or full-length Sororin.

Response: We thank the reviewer for this suggestion. As requested, we tested whether SMC3 acetylation influences the Sororin interaction. New data show that neither acetylation-deficient (K105A/K106A) nor acetylation-mimetic (K105Q/K106Q) mutations in SMC3 affected co-immunoprecipitation of Sororin-GFP with the SMC3-HD-Flag/RAD21-N103-Myc complex (Supplementary Fig. 6G; **new data**).

Consistently, AlphaFold3 modeling indicates that K105 and K106 are spatially distant from the Sororin-binding residues (K1034, K1038, R1099) on SMC3 (Supplementary Fig. 6H; **new data**). These findings indicate that SMC3 acetylation licenses Sororin recruitment without directly regulating its binding to the RAD21-SMC3 interface.

2. In Figure 2 (G, H), tethering Sororin-CTR to centromeres had little effect on cohesion in cells co-depleted of Esco1/2. Does Sororin-CTR affect its binding to chromatin at the cohesion establishment stage? It would be interesting to test whether expression of H2B-fused Sororin-CTR can rescue cohesion defects in Esco1/2-depleted cells.

Response: We appreciate this suggestion. New experiments demonstrate that co-depletion of Esco1/2 causes comparable cohesion defects in both H2B-GFP and H2B-Sororin-CTR-GFP cells (new Supplementary Fig. 3C-F; **new data**), demonstrating that Esco1/2 perform non-redundant functions that cannot be bypassed by Sororin.

3. Sororin is phosphorylated by mitotic kinases PLK1, CDK1, and Aurora B in vivo, which mediates its degradation. The authors found that Aurora B phosphorylation of Sororin at S145 reduces its interaction with Pds5 in vitro. Is this the case in vivo? What's the order of these phosphorylation events which may mediate dissociation from PDS5 and degradation sequentially?

Response: We have addressed the reviewer's questions as follows:

1) New co-immunoprecipitation data confirm that the phospho-mimetic Sororin-S145E mutant fails to interact with Myc-Pds5B in G2-synchronized cells (Supplementary Fig. 5D; **new data**);

2) Aurora B inhibition in mitotic cells enhanced Pds5B binding to wild-type Sororin-GFP, but not to Sororin-S145E-GFP (Supplementary Fig. 5E; **new data**). This confirms that Aurora B phosphorylation at S145 disrupts the Sororin-Pds5 interaction *in vivo*.

Additionally, as the reviewer notes, literature indicates Sororin is phosphorylated in prophase/prometaphase (causing Pds5 dissociation) and subsequently degraded in anaphase.

4. Sgo1 plays a well-established role in sustaining Sororin activity at mitotic centromeres. Figure S2 (A-C) shows that tethering Sororin-CTR to centromeres, as a fusion protein with a CENP-B fragment, only partially restored cohesion defects caused by Sgo1 depletion. This is somewhat striking, since Sgo1 is generally regarded as a master protector of centromeric cohesion. This result suggests that Sgo1 may have an additional role in protecting cohesion along mitotic chromosome arms. To test this possibility, the authors should assess the effect of chromatin-tethered Sororin-CTR on cohesion in Sgo1-depleted cells.

Response: Thanks for this excellent point. New data show that Sgo1 knockdown caused severe cohesion loss in H2B-GFP cells, but this defect was largely suppressed in H2B-Sororin-CTR-GFP cells (Supplementary Fig. 3A, B; **new data**). This indicates that chromatin-tethered Sororin-CTR can largely bypass the requirement for Sgo1, supporting the model that Sgo1 protects cohesin both at centromeres and on chromosome arms.

5. Figure 3H shows that, compared to H2B-GFP-expressing cells, the frequency of anaphases with chromosome bridges was strongly increased in H2B-Sororin-CTR-expressing cells, whereas the lagging chromosome rates remained comparable. Given the critical role of centromeric Aurora B kinase in correcting erroneous kinetochore-microtubule attachments, the authors should compare Aurora B accumulation at mitotic centromeres in cells expressing H2B-GFP versus H2B-Sororin-CTR-GFP.

Response: We have added the requested analysis. New data show that centromeric accumulation of Aurora B, which is important for correcting erroneous kinetochore-microtubule attachments, was comparable in cells expressing H2B-GFP and H2B-Sororin-CTR-GFP (Supplementary Fig. 3G, H; **new data**), indicating that the increased chromosome bridges do not result from Aurora B mislocalization.

6. Figures 4G and S3 show that GST-Sororin-CTR interacted with SMC3-HD only when it formed a subcomplex with the N-terminal fragment of RAD21. This suggests that Sororin-CTR binds the cohesin core complex in which the DNA-exit gate (i.e, the SMC3-HD/RAD21-N interface) is closed, but not when the gate is open. Because cohesin-mediated sister chromatid cohesion is established during S phase, maintained in G2, and largely lost in mitosis, the authors should compare the relative binding

efficiency of Sororin-CTR to endogenous cohesin core complex in early S phase versus G2, and in G2 versus mitosis.

Response: We have performed the suggested experiments. New data demonstrate that GST-Sororin-CTR bound endogenous SMC3, SMC1, and RAD21 more efficiently in G2-phase cell lysates than in early S-phase lysates (Fig. 4K; **new data**). Furthermore, this interaction persisted in late-G2 phases but was markedly reduced in mitotic cells (Fig. 4L; **new data**). Thus, Sororin-CTR binding to cohesin peaks during the cohesion-maintenance phase of the cell cycle.

7. Using bacterially expressed and purified proteins, Figure 4H shows that GST-Sororin-CTR pulled down the complex formed between RAD21-N103 and SMC3-HD. This indicates a direct interaction between Sororin-CTR and the RAD21/SMC3-HD interface. To corroborate this, the authors should test whether the RAD21-N103/SMC3-HD complex can directly pull down GST-fused full-length Sororin.

Response: We have added this corroborating experiment. Using purified recombinant proteins, we show that the RAD21-N103/SMC3-HD complex directly pulls down GST-fused full-length Sororin (Fig. 4J; **new data**). This confirms that full-length Sororin directly engages the RAD21-SMC3 interface.

8. It is well known that Sororin phosphorylation during mitosis promotes cohesin release from chromosome arms. Figure 5A shows that Sororin efficiently co-immunoprecipitated RAD21-N103, SMC3-HD, and endogenous Pds5B in G2-phase cell. In mitosis, Sororin binding to RAD21-N103 and SMC3-HD persisted, while the Sororin-Pds5B interaction was disrupted. Figure 5B demonstrates that co-inhibition of Cdk/Plk1/Aurora kinase activity during mitosis restored the Sororin-Pds5B interaction but did not affect Sororin binding to RAD21-N103/SMC3-HD. These results imply that Sororin-CTR binding to the RAD21/SMC3-HD interface is insensitive to mitotic phosphorylation, in contrast to the Sororin-Pds5B interaction. To strengthen this conclusion, the authors should examine the effect of treating mitotic cell lysates with lambda-phosphatase on Sororin binding to RAD21-N103/SMC3-HD and Pds5B.

Response: As suggested, we treated nocodazole-treated HEK-293T cell lysates with λ -phosphatase. New data show this treatment markedly enhances the Sororin-Pds5B co-immunoprecipitation, without affecting interactions with RAD21-N103 or SMC3-HD (Fig. 5B; **new data**). This strengthens our conclusion that the Sororin-Pds5 interaction is phosphorylation-sensitive, while Sororin's binding to the RAD21-SMC3 interface is not.

9. Figure 6 (C, D) shows simultaneous mutation of W233 and E252 to alanine and lysine, respectively disrupts Sororin-CTR binding to the SMC3-HD/RAD21-N103 subcomplex and to endogenous cohesin. A Sororin-E252Q mutation is reported in the cancer genome database. It would be interesting to see to what extent this mutation can affect the binding of full-length Sororin to SMC3 and RAD21 in cells.

Response: We tested a cancer-associated Sororin-E252Q mutation. New data show this mutation reduces the co-precipitation of SMC3 and RAD21 with full-length

Sororin-Flag in cells (Supplementary Fig. 6C; new data), suggesting that this mutation partially compromises Sororin's gate-locking function and may contribute to genomic instability.

Minor points:

1. The statistical results lack critical information: such as what the values represent (\pm SD or \pm SEM), the method of significance analysis, the corresponding p-values, and the number of biological replicates. Fig. 1D and 1G; Fig. 2D and 2E lack significance analysis. Fig. 3G lacks error bars, and no significance analysis was performed for Fig. 3H and 3I. Fig. 7B, 7G, 7I; Fig. 8I; Fig. 9B, 9E lack significance analysis.

Response: We have updated all figures to include error bars (\pm standard deviation), specified statistical tests (unpaired two-tailed t-test), provided exact p-values, and indicated the number of biological replicates (n).

2. The Methods section states that statistical analyses were performed using GraphPad Prism version 6, but do not specify the exact tests used (e.g., unpaired t-test, ANOVA). This should be clarified.

Response: The Methods section now specifies the exact statistical tests used for each analysis.

3. Could the authors provide the 3D structure of the cohesin complex or protein-protein interactions predicted by AlphaFold3?

Response: An AlphaFold3-predicted 3D structure of a Sororin-CTR peptide binding to the interface between RAD21-NHD and SMC3-HD is now provided as Supplementary Data 2.

4. Some grammars and typos, for instance: "phenocopying Wapl loss" in the Abstract better to be changed to "phenocopying the effects of Wapl loss". "Dysregulation of cohesin underlie developmental disorders and contribute to tumorigenesis" in the Introduction section should be changed to "Dysregulation of cohesin underlies developmental disorders and contributes to tumorigenesis". "Wapl regulates the dynamic association of cohesin with chromatin" in the Results section better to be changed to "Wapl regulates the dynamics of cohesin's association with chromatin". "To rule out the possibility that the failure of Sororin (1-149)-GFP and Sororin (1-222)-GFP to promote cohesion was due to deficient chromatin binding" in the Results section better to be changed to "To rule out the possibility that the failure of Sororin (1-149)-GFP and Sororin (1-222)-GFP to promote cohesion was due to defective chromatin binding".

Response: All grammatical errors and typos have been corrected.

We thank the reviewer again for their constructive feedback, which has significantly strengthened the manuscript.

Reviewer #3 (Remarks to the Author):

The manuscript by Chen et al. investigates the mechanisms by which Sororin functions to stabilize cohesive cohesin on DNA. While Sororin's antagonistic role against WAPL-mediated cohesin removal has been recognized, the underlying mechanisms remained poorly defined. Previous models proposed that Sororin and WAPL compete for PDS5 binding, however, this study reveals a different mechanism in which, in addition to interacting with PDS5, Sororin directly interacts with the cohesin complex, functioning as a gatekeeper for the SMC3-RAD21 exit gate. The authors demonstrate that the C-terminal domain of Sororin is sufficient to maintain sister chromatid cohesion and that tethering of a C-terminal Sororin construct to centromeres functionally protects sister chromatid cohesion and counters WAPL-mediated removal. Despite this protective effect, knockdown of ESCO1/2 still results in cohesion defects even when Sororin is tethered to centromeres, indicating that cohesin acetylation occurs upstream of Sororin binding and that stabilization of Cohesin by acetylation is required independently to maintain cohesion. Importantly, when Sororin is tethered along the chromosome arms and cannot be removed, it leads to mitotic segregation defects and abnormal chromosome compaction, indicating the importance of dynamic regulation and maintenance of cohesive cohesin at centromeres but not at chromosome arms in mitosis. Through immunoprecipitation experiments and elegant structure-function analyses, the authors demonstrate that Sororin directly interacts with cohesin and define the specific interaction interfaces. Using both in vitro and in vivo binding assays, they map the binding interfaces at the SMC3-RAD21 junction and validate these through reciprocal mutagenesis. Critically, co-depletion of both WAPL and Sororin followed by expression of binding mutants restores cohesion, confirming that exit gate locking is required specifically to block WAPL-mediated unlocking rather than to maintain cohesion directly.

In addition to its interaction with cohesin, the authors find that Sororin binds PDS5B in an Aurora phosphorylation-dependent manner, identifying the specific phosphorylation residue and testing this through both in vitro and cell-based kinase assays. Interestingly, the phosphorylation of Sororin by Aurora regulates Sororin binding to PDS5B but not to cohesin, indicating that the association of Sororin with cohesin and PDS5 is independent, although the Sororin-cohesin interaction was reduced.

Together, this work provides a comprehensive answer to the long-standing question of how Sororin counteracts WAPL-mediated cohesin removal. The authors employ multiple complementary approaches to test their model and draw balanced conclusions from their data. Given its relevance to genome regulation, human health, and its potential to advance the cohesin field, I recommend this manuscript for publication in Nature Communications. I have listed a few minor suggestions below that could strengthen the manuscript.

Response: We thank the reviewer for the supportive assessment and insightful suggestions, which have helped us refine our model and strengthen the manuscript.

Below are our responses to their specific points.

Suggestions:

1. The authors propose a mechanism in which phosphorylation of Sororin by AURKB releases the Sororin-PDS5 interaction while maintaining the Sororin-cohesin interaction, providing a switch whereby arm cohesin (but not centromeric cohesin) can be released during early mitosis. They suggest that arm cohesin has less Sororin CTR engagement due to lowered Sororin levels, allowing for exit gate opening. However, I find this model somewhat confusing: if Sororin is still interacting with PDS5 despite lowered Sororin levels, why would the CTR of these same Sororin proteins not also bind to cohesin in these same complexes? An alternative interpretation is that the pull-down of Sororin with cohesin in the ASE mutants reflects Sororin bound to soluble cohesin that is being removed from chromatin, rather than chromatin-bound cohesive cohesin. Typically, Sororin only interacts with chromatin-bound cohesin except in cases where cohesin cannot be acetylated; however, it is possible the authors are capturing this transient interaction when cohesin is being released. To help support the authors' proposed model, it would be useful to perform the IP experiments coupled with cellular fractionation. Additionally, it would be useful to include chromosome spreads in the ASE mutant in the absence of endogenous Sororin to show that cohesive cohesin remains intact specifically at centromeres but not at chromosome arms. However, it is appreciated that these are complicated experiments, and it would be appropriate to alternatively mention other possible hypotheses in the discussion.

Response: We thank the reviewer for this thoughtful suggestion, which prompted us to further test our model. To distinguish chromatin-bound from soluble cohesin pools, we performed chromatin fractionation followed by co-immunoprecipitation. Sororin-GFP stably expressed in HeLa cells co-precipitates RAD21, SMC3, and Pds5B specifically from the chromatin-bound fraction of mitotic cells, but not from the soluble fraction (Fig. 4M, N; **new data**). This confirms that the Sororin-cohesin interactions we detect are primarily associated with chromatin-bound cohesin, not with a soluble pool being removed. This finding supports our proposed model wherein centromeric Sororin remains stably engaged with chromatin-bound cohesin via both its Pds5-binding motif and CTR, while on chromosome arms, Aurora B phosphorylation promotes its dissociation from Pds5 and cohesin release.

2. The authors find that tethering of Sororin to centromeres can only partially rescue Shugoshin knockdown, consistent with a model in which Shugoshin-mediated PP2A dephosphorylation blocks the release of the Sororin-PDS5 interaction at centromeres, indicating both interaction with PDS5 and cohesin is required to protect cohesin from premature removal. To further test this regulatory pathway, can a non-phosphorylatable Sororin mutant bypass the requirement for Shugoshin protection, functioning to block Aurora phosphorylation? This experiment would provide additional validation of the proposed phosphorylation-dependent switch mechanism and the need for both sororin interaction interfaces to mediate cohesive cohesin.

Response: We performed the suggested experiment. In mitotic cells, Sororin-S145N exhibited enhances association with Pds5B, SMC3, and RAD21 compared with wild-type Sororin, whereas no difference is observed in G2-phase cells (Supplementary Fig. 5F, G; **new data**). Because Sgo1-associated PP2A protects centromeric cohesion by dephosphorylating Sororin, we examined whether the S145N mutant could bypass Sgo1 depletion. Although the S145N mutant partially rescued the cohesion defect caused by Sgo1 knockdown, it does not fully restore cohesion (Supplementary Fig. 5H-J; **new data**). This indicates that preservation of the Sororin-Pds5 interaction via S145 dephosphorylation represents a major, but not exclusive, function of Sgo1-PP2A in cohesion protection. We have incorporated this nuance into the revised manuscript.

3. It would be useful to add statistics to the figures throughout.

Response: We have added statistics to the figures throughout.

4. Typo line 129, “whether selectively retention”.

Response: All typos have been corrected.

Reviewer #4 (Remarks to the Author):

How sister chromatid cohesion is established and maintained is a key question in the field of chromosome segregation. Extensive efforts from many labs have identified the major players in these processes and revealed a lot of basics about how these players work together to regulate sister chromatid cohesion. Among these players are Wapl, Pds5 and Sororin. It has been believed that the positive factor Sororin competes against factor Wapl for Pds5 binding, thus dictating “on and off” of sister chromatid cohesion. It has also been shown that the C-terminal domain of Sororin is required for sister chromatid cohesion; but the underlying mechanism is unknown. In addition, it remains elusive how these proposed mechanisms work together to regulate sister chromatid cohesion. Very interestingly, Chen et.al in this study found that the competition between Wapl and Sororin on Pds5 may not be that important for the regulation of sister chromatid cohesion. Instead, they revealed a novel mechanism, in which Sororin engage its C-terminal domain to directly binds to “the gate” of the cohesin complex that forms between Smc3 and Rad21, which competes against the action of Wapl, thus protecting sister chromatid cohesion. The major conclusions in this paper are solid and largely supported by plenty of data. These conclusions will contribute for us to further understand the molecular mechanisms of sister chromatid cohesion. The experiments were well controlled and carried out. And the paper was well organized and written. Especially, a series of point mutants in cohesin and Sororin identified in this study significantly strengthen their conclusion. I congratulate the authors for this great study. I do not have any major concerns and support the publication in NC.

Response: We thank the reviewer for the positive and insightful evaluation of our manuscript. We are grateful for the acknowledgment that our work reveals a novel

mechanistic role for Sororin's C-terminal domain and solidifies our understanding of cohesion maintenance.

The reviewer correctly identifies the key contributions of our study: moving beyond the established model of Sororin-Wapl competition for Pds5 binding, and instead providing direct evidence that Sororin acts as a molecular lock at the SMC3-RAD21 exit gate to antagonize Wapl and protect cohesion. We are pleased that the reviewer found the conclusions solid, the data plentiful and well-controlled, and the use of precise point mutants to be particularly strengthening.

Minor points:

1. The authors already demonstrated that the expression of the two C-terminal domains are able to close the chromosome arm. Especially, for the smaller C-terminal domain (223-252), I believe that most of regulatory domains or residues in Sororin have been deleted out, for example, the Cdk1 and Aurora B phosphorylation sites. Whether this fragment will always bind Smc3/Scc1 throughout the cell cycle? Does it also generate anaphase bridges just like the H2B-fused fragment? It would be nice if author have the data. Otherwise, a brief discussion would also be fine.

Response: We appreciate this excellent question regarding the behavior of the minimal Sororin-CTR fragment. We have addressed this point with new experiments examining its cell cycle-dependent binding and functional consequences.

1) New pull-down assays reveal that GST-Sororin-CTR bound endogenous SMC3, SMC1, and RAD21 more efficiently in G2-phase cell lysates than in early S-phase lysates (Fig. 4K; **new data**). Furthermore, this interaction persists in late-G2 phases but is markedly reduced in mitotic cells (Fig. 4L; **new data**). Thus, Sororin-CTR binding to cohesin peaks during the cohesion-maintenance phase of the cell cycle.

2) As the reviewer hypothesized, cells overexpressing the Sororin-CTR-GFP fragment (lacking N-terminal regulatory domains) also exhibit a significant increase in anaphase chromosome bridges (Supplementary Fig. 3I-K; **new data**), linking these defects directly to the CTR's cohesion-stabilizing activity.

These data collectively support the model that the Sororin CTR functions as a critical, cell cycle-regulated effector whose engagement with the RAD21-SMC3 interface is sufficient to stabilize cohesin and whose dysregulated retention causes segregation defects.

Reviewer #5 (Remarks to the Author):

Response: We thank the reviewer for the contribution to the peer review process and for the co-assessment of our manuscript.

Point-to-point response to reviewer comments

REVIEWER COMMENTS

Reviewer #1 (Remarks to the Author):

The authors have addressed my concerns with significant new data that substantially improve the rigor of the manuscript. I have a few general comments. The figure legends require further improvement so that readers can fully appreciate and interpret the data without having to refer back to the Results section. For example, in Fig. 4M (top two panels), “Sororin-GFP,” “S.” and “L.” should be explicitly defined in the legend rather than left to assumption.

Response: We thank the reviewer for the positive assessment and for highlighting this important point regarding clarity. We have now thoroughly revised all figure legends throughout the manuscript to ensure they are comprehensive and self-sufficient. Specifically, for Figure 4M, N, we have amended the legend to explicitly state: “HeLa cells stably expressing Sororin-GFP were synchronized in mitosis by nocodazole treatment for 16 h and subjected to chromatin fractionation. A portion of the whole cell lysate (WCL), supernatant, and chromatin pellet was analyzed by immunoblotting for histone H3, GAPDH, GFP, RAD21, SMC3, and Pds5B (M). The remaining chromatin pellet and 5% of the supernatant were subjected to immunoprecipitation with anti-GFP beads, followed by immunoblotting for GFP, RAD21, SMC3, and Pds5B (N). S., short exposure; L., long exposure.”

This revision has been applied consistently across all figures to enhance reader comprehension without necessitating referral to the main text.

Reviewer #2 (Remarks to the Author):

All comments have been addressed with new data. It's now solid and ready for publication.

Response: We thank the reviewer for the positive and conclusive evaluation of our revised work.

Reviewer #3 (Remarks to the Author):

The manuscript by Chen et al. addresses a long standing question of how Sororin acts to counteract WAPL based cohesin removal for the establishment of sister chromatid cohesion. In this new revised version of the manuscript the authors have added a number of experiments, included new analysis and statistics, and importantly added in additional discussion points that together has significantly strengthened the work. With these changes all of our concerns have been addressed and we would

recommend this manuscript for publication in Nature Communications in its current form.

Response: We are grateful to the reviewer for recognizing the significance of our study and for the encouraging summary. We appreciate their affirmation that the new experiments, analyses, and discussions have satisfactorily addressed their previous concerns.

Reviewer #4 (Remarks to the Author):

I am satisfied with the authors' response.

Response: We thank the reviewer for the time and for confirming their satisfaction.

Reviewer #5 (Remarks to the Author):

Response: We acknowledge the contribution of the early-career researcher and thank them for their participation in the review process.